# Differentiation of Myocardial Properties in Physiological Athletic Cardiac Remodeling and Mild Hypertrophic Cardiomyopathy

**DOI:** 10.3390/biomedicines12020420

**Published:** 2024-02-12

**Authors:** Lars G. Klaeboe, Øyvind H. Lie, Pål H. Brekke, Gerhard Bosse, Einar Hopp, Kristina H. Haugaa, Thor Edvardsen

**Affiliations:** 1Precision Health Center for Optimized Cardiac Care (ProCardio), Department of Cardiology, Oslo University Hospital, Rikshospitalet, 0424 Oslo, Norway; lgklaeboe@yahoo.no (L.G.K.); oyvlie@gmail.com (Ø.H.L.); kristina.haugaa@medisin.uio.no (K.H.H.); 2Division of Radiology and Nuclear Medicine, Oslo University Hospital, Rikshospitalet, 0424 Oslo, Norway; gbosse@ous-hf.no (G.B.); ehopp@ous-hf.no (E.H.); 3Faculty of Medicine, University of Oslo, 0316 Oslo, Norway; 4KG Jebsen Cardiac Research Centre, Institute of Clinical Medicine, Faculty of Medicine, University of Oslo, 0316 Oslo, Norway

**Keywords:** athletes’ hearts, cardiac magnetic resonance imaging, hypertrophic cardiomyopathy, strain imaging

## Abstract

Clinical differentiation between athletes’ hearts and those with hypertrophic cardiomyopathy (HCM) can be challenging. We aimed to explore the role of speckle tracking echocardiography (STE) and cardiac magnetic resonance imaging (CMR) in the differentiation between athletes’ hearts and those with mild HCM. We compared 30 competitive endurance elite athletes (7% female, age 41 ± 9 years) and 20 mild phenotypic mutation-positive HCM carriers (15% female, age 51 ± 12 years) with left ventricular wall thickness 13 ± 1 mm. Mechanical dispersion (MD) was assessed by means of STE. Native T1-time and extracellular volume (ECV) were assessed by means of CMR. MD was higher in HCM mutation carriers than in athletes (54 ± 16 ms vs. 40 ± 11 ms, *p* = 0.001). Athletes had a lower native T1-time (1204 (IQR 1191, 1234) ms vs. 1265 (IQR 1255, 1312) ms, *p* < 0.001) and lower ECV (22.7 ± 3.2% vs. 25.6 ± 4.1%, *p* = 0.01). MD > 44 ms optimally discriminated between athletes and HCM mutation carriers (AUC 0.78, 95% CI 0.65–0.91). Among the CMR parameters, the native T1-time had the best discriminatory ability, identifying all HCM mutation carriers (100% sensitivity) with a specificity of 75% (AUC 0.83, 95% CI 0.71–0.96) using a native T1-time > 1230 ms as the cutoff. STE and CMR tissue characterization may be tools that can differentiate athletes’ hearts from those with mild HCM.

## 1. Introduction

The symmetrical chamber dilation, increased myocardial mass and relative bradycardia observed in athletes’ heart [1] are easily distinguished from the typical morphological and functional features of severe hypertrophic cardiomyopathy (HCM). The differentiation between benign athletic cardiac remodeling and mild phenotypic HCM may, however, be a more difficult task [2,3]. Furthermore, the prevalence of HCM is high in the general population, and it will obviously be present in some individuals performing sports. ECG changes may not be conclusive [4], and conventional echocardiography and cardiac magnetic resonance imaging (CMR) have important shortcomings [5]. Genetic tests have similar limitations, as 40% of HCM patients do not carry known pathogenic mutations [6], and the identification of genetic mutations of uncertain significance imposes further uncertainty on the well-being of an athlete and their immediate family.

Myocardial alterations in typical HCM, including myocardial disarray and fibrosis, induce heterogeneity in ventricular contractions not observed in healthy athletes [7]. Strain echocardiography has been reported to be promising in the differentiation between typical phenotypic HCM and athletes’ hearts [7,8,9], but has not been evaluated in the discrimination of mild HCM from athletic cardiac remodeling. CMR studies have reported high intracellular volumes in the physiological hypertrophy of athletes’ hearts [10], and conversely high extracellular volumes (ECVs) in patients with HCM [11]. However, no head-to-head comparisons between athletes and mutation-positive HCM with a mild phenotype using the native T1-time on the same magnet have been performed previously. 

We aimed to describe mutation-positive HCM with a mild phenotype and athletes’ hearts using strain echocardiography and CMR indices in order to evaluate their role in the differentiation between the two entities.

## 2. Materials and Methods

### 2.1. Study Population

Study participants were recruited between April 2016 and June 2017 in a cross-sectional study. We informed high-level cycling and cross-country skiing teams and other endurance athletes in Oslo about the planned study. Healthy volunteering athletes were encouraged to contact our study group to be considered for inclusion, as previously described [12]. A selection of clinical data, two-dimensional (2D) echocardiographic measurements and conventional CMR indices from these healthy volunteering athletes have been previously published by Lie et al. in a comparison with athletes with ventricular arrhythmias [12]. 

Athletes disclosed their exercise history from school age to present in a structured interview. Exercise was reported as the type of activity/sport, graded at perceived intensity levels 1–3 (light, moderate, vigorous), and the duration was reported as hours per week, months per year, and years. The intensity of the reported activities was rated according to the Compendium of Physical Activities and quantified as metabolic equivalents (METs) [13]. Exercise dose, expressed as MET-hours, was estimated by means of the multiplication of exercise intensity and exercise duration [14]. Only mutation-positive, otherwise healthy, sedentary HCM patients with mild phenotypes and a maximal wall thickness of 12–16 mm were included for comparison. These patients were identified by a cascade genetic screening program of 144 HCM probands. Genetic testing was performed as previously described [15]. Only patients with pathogenic or likely pathogenic mutations were included (myosin binding protein C 3 (*MYBPC3*, n = 13), beta-myosin heavy chain 7 (*MYH7*, n = 4), troponin I (*TNNI3*, n = 2) and troponin T (*TNNT2*, n = 1)). 

Subjects with more than moderate valvular disease, left ventricular (LV) outflow track obstruction, atrial fibrillation, hypertension, coronary artery disease, diabetes mellitus, a pacemaker/implantable cardioverter defibrillator or reported use of performance-enhancing drugs were excluded. The study was approved by the Regional Committee for Medical Research in Norway (approval number 2015/1593). Written informed consent was obtained from all study participants.

### 2.2. Echocardiography

All subjects underwent transthoracic echocardiographic examinations using the Vivid 95 ultrasound system (GE Vingmed Ultrasound AS, Horten, Norway). Data were analyzed with EchoPAC version 201 software (GE Vingmed Ultrasound AS, Horten, Norway). Echocardiographic parameters were assessed in agreement with the expert consensus document on multi-modality imaging approaches to athletes’ hearts [16]. Maximal wall thickness was assessed by means of 2D echocardiography. Full-volume three-dimensional (3D) LV acquisitions were obtained in the left decubital position during a complete breath-hold for six heart cycles, achieving frame rates of >35 frames per second. Three-dimensional LV volumes, mass and ejection fraction (EF) were estimated during post-processing with the LVQ-tool (GE Healthcare). The left atrial volume was calculated using the biplane area–length method. 

The LV global longitudinal strain (GLS) was derived from speckle tracking analyses on 2D gray scale image loops with >50 frames per second from the three apical views, and was expressed as the average peak systolic strain in a 16 segment LV model [17]. LV mechanical dispersion was defined as the standard deviation of time from Q/R on the surface ECG to the peak negative strain in 16 LV segments [18] (Figure 1). 

### 2.3. Cardiac Magnetic Resonance Imaging

CMR was performed on a single 3 Tesla unit (Philips Ingenia, Philips Healthcare, Best, The Netherlands). Cine sequences of standardized long-axis projections and multiple short-axis projections covering both ventricles were performed (Figure 2). LV and right ventricular (RV) volumes were calculated semiautomatically by experienced radiologists using the freely available software Segment v2.2 R6405 [19]. After intravenous injection of the contrast medium, late gadolinium enhancement (LGE) was assessed in the steady state and qualitatively recorded as being present or absent. Native T1 time was recorded as the mean septal value. The ECV was calculated as the ratio between myocardial and blood relaxivity change after contrast medium injection, multiplied by the blood ECV (1—hematocrit) and expressed as a percentage of the total myocardium. Hematocrit values were obtained immediately prior to investigation.

### 2.4. Statistics

Values were presented as mean ± standard deviations, frequencies with percentages, and medians with the interquartile range (IQR) and were compared using Student’s *t*-test, *χ*^2^, Fischer’s exact test or the Mann–Whitney U test as appropriate (SPSS statistics 27.0, SPSS Inc., Chicago, IL, USA), The discriminatory ability of imaging markers from strain echocardiography and CMR were assessed using receiver operator characteristics (ROC) curves and expressed by the area under the curve (AUC) with a 95% confidence interval (CI). The coordinates from the ROC curves closest to the upper left corner defined the optimal cut-off values. Two-sided *p*-values < 0.05 were considered statistically significant.

## 3. Results

### 3.1. Clinical Characteristics 

We included 20 healthy, sedentary mutation-positive HCM subjects with a mild phenotype (15% female, age 51 ± 12 years) identified by means of cascade genetic screening of a population of 144 family members (Table 1). Thirty healthy competitive endurance elite and elite master athletes were used as a control group and included for comparison [12]. These athletes (7% female, age 41 ± 9 years) had an accumulated life-time exercise dose of 94 (IQR 64–154) thousand MET-hours. The represented sports were cycling (63%), cross-country skiing (27%), rowing (3%) and triathlon (3%). The athletes were younger and had a lower body surface area (Table 1). 

### 3.2. Cardiac Imaging

The maximal wall thickness was 10 ± 2 mm in athletes and 13 ± 1 mm in mild phenotypic HCM, as measured by 2D echocardiography (*p* < 0.001). Table 1 shows that athletes and mutation positive HCM subjects had a similar LV mass and LV EF, but the LV volumes established by 3D echocardiography were greater in athletes (*p* < 0.001). Furthermore, the RV function assessed by means of 2D fractional area change (FAC) was better in mutation-positive HCM subjects (*p* = 0.001). The left atrium was similarly enlarged in mild HCM subjects and athletes. Other indices of diastolic function (E/A ratio, e’ and E/e’ ratio) were normal in both groups, although the values of these diastolic variables were higher in athletes (Table 1). Speckle tracking echocardiography showed that the athletes and the mutation-positive HCM subjects had similar, normal GLS, but mechanical dispersion was more pronounced in mutation-positive HCM subjects (*p* = 0.001). 

CMR was completed in 19 (86%) mild phenotypic HCM subjects and 29 (97%) healthy athletes. Higher LVEF and RVEF were found in mild HCM subjects, while athletes had greater LV and RV volumes. LGE was more prevalent in mild phenotypic HCM than in healthy athletes. Athletes had a lower native T1 time and a lower ECV (Table 1). 

Figure 3 shows the ability of imaging markers from strain echocardiography and T1 mapping techniques to identify HCM mutation carriers from athletes in the study population evaluated via ROC analysis. These results are summarized in Table 2. According to AUC, the native T1 time and mechanical dispersion were the two imaging markers with the most promising discriminatory ability, while GLS showed poor discriminatory ability. The ROC curves show that the native T1 time established by CMR may identify HCM mutation carriers with excellent sensitivity and good specificity (Figure 3). The optimal discrimination value of native T1 time in this study population was >1230 ms. This cut off identified all HCM mutation carriers (100% sensitivity) with a specificity of 75%. LV mechanical dispersion > 44 ms optimally discriminated between athletes and HCM mutation carriers (AUC 0.78, 95% CI 0.65–0.91). The AUC for the combined ROC curve for native T1 time and mechanical dispersion was 0.82 (95% CI 0.70–0.95), suggesting no added discriminatory value when combining the two imaging indices from CMR and strain echocardiography with the highest AUC.

In addition to the main parameters of interest (mechanical dispersion, GLS, native T1 time and ECV), we also present ROC analysis of imaging indices that were shown to be significantly different among athletes and HCM mutation carriers in Appendix A, although these parameters were not the primary target of investigation in this study.

## 4. Discussion

In the current study, mutation-positive, mild phenotypic HCM patients and competitive endurance athletes underwent evaluation via echocardiography and subsequent CMR in a comparative experimental setting. Sedentary subjects with mutation-positive mild HCM had a more pronounced mechanical dispersion, a higher native T1 time, a higher ECV and a higher prevalence of LGE, suggestive of greater disarray and fibrosis than athletes. Among the strain parameters, GLS could not differentiate mild HCM from athletes’ hearts, while mechanical dispersion showed promising discriminatory ability. The native T1 time established by CMR showed better diagnostic performance than ECV and mechanical dispersion in identifying HCM mutation carriers in a mixed population with endurance athletes. 

### 4.1. Physiological and Pathological Myocardial Remodeling

Undiagnosed underlying heart disease is the leading cause of sudden cardiac death in athletes. In younger athletes, inherited cardiomyopathies such as HCM are the main causes of these tragic events [20]. Traditionally, people with HCM have been advised not to participate in sports. This view has become more nuanced in recent years as data regarding athletic cardiac remodeling and the impact of physical activity in HCM are accumulating [21,22]. HCM and athletes’ hearts share common morphological features that might complicate the differentiation between the two clinical entities. A common scenario is when maximal wall thickness falls into the “gray zone” between the pathological hypertrophy of HCM and physiological athletic remodeling, drawing attention to a possible underlying inherited cardiomyopathy. Genetic testing may be helpful, although a large proportion of patients with HCM belong to a negative genotype, and the pathogenicity of a genetic mutation may be difficult to establish [23]. The fact that patients with an HCM-causing genetic mutation may present with virtually any wall thickness complicates this further [24]. New biomarkers that clarify the patient’s condition are therefore of interest. The current study focused on strain echocardiography, which is widely available, and the more advanced CMR-derived native T1 time and ECV, in a comparative study to evaluate whether these indices could demonstrate differences related to myocardial properties in high-performance endurance athletes and patients with mild HCM without classical phenotypical features or discernable comorbidities that could cause cardiac remodeling.

### 4.2. Imaging Myocardial Properties

Myocardial cellular disarray and fibrosis are common structural alterations in HCM [25]. These changes may even be found in young HCM patients who are completely asymptomatic until the event of sudden cardiac death [26]. Using imaging techniques that can accurately assess myocardial function or characterize tissue may detect the disease at an earlier stage. 

Strain echocardiography offers a sensitive functional assessment of the myocardium. The rationale for using strain to evaluate early pathological remodeling is the ability to uncover subtle changes in myocardial function that are not detected by conventional echocardiographic measures. Focusing on functional measures as a surrogate for altered myocardial properties rather than tissue characteristics, strain has only moderate diagnostic performance compared to CMR to detect fibrosis [27].

Abnormal GLS has been suggested as a diagnostic tool to distinguish HCM from athletic cardiac remodeling [8,9], but one previous report found no discriminatory value for GLS [28]. Without reporting the genotype, the same report described mechanical dispersion to be promising in differentiating athletes from a population of patients with a typical HCM phenotype. In the present study, pronounced mechanical dispersion revealed LV contraction heterogeneity even in milder HCM disease, although other measures of systolic LV function were similar in the athletes. In accordance with a previous report, we found no significant differences in longitudinal function by means of GLS between athletes and patients with HCM [28].

CMR using LGE can identify the patchy replacement fibrosis of HCM typically present in the most maximally hypertrophied segments [5]. However, as approximately 50% of patients have no LGE, HCM cannot be ruled out based on LGE assessment [5]. Therefore, the novel techniques of assessing diffuse myocardial fibrosis according to the native T1 time and ECV have gained more attention. In our cohort, we found more individuals with LGE indicative of replacement fibrosis in the mild HCM phenotype. A longer native T1 time and a higher ECV were consistent with different tissue characteristics in mild HCM, possibly due to more cellular disarray and cellular matrix expansion compared to athletes. This provides a plausible explanation for the more pronounced mechanical dispersion observed in the HCM group, as changes in myocardial composition have previously been linked to heterogeneous ventricular contractions in HCM [7]. 

### 4.3. Diagnostic Considerations and Future Perspectives

As illustrated by the ROC curves, sensitivity and specificity vary among echocardiographic and CMR indices. The current study preselected athletes and mild phenotypic HCM patients under rigorous circumstances using genetic testing of known mutations as a gold standard to identify HCM in a mixed population with athletes. 

ROC analysis suggests that the native T1 time has better diagnostic performance than mechanical dispersion and the ECV, and that the discriminatory ability of GLS is inadequate to separate athletes and HCM. Moreover, combined ROC analysis indicates no additional discriminatory improvement compared to a multimodality approach combining STE and CMR, but this may be due to the limited sample size. Although the discriminatory ability of strain imaging and T1 mapping techniques may be promising, the clinical cut off values or usefulness cannot be inferred from this study design. The next logical step would be an investigation evaluating the same imaging methods in a clinical setting in athletes with suspected HCM, where the discrimination properties of the discussed methods are likely to change.

This study focused on parameters from strain imaging and CMR. There were, however, indications of discriminatory ability among more conventional imaging parameters (Appendix A). Worthy of note is that E/e’ had one of the highest AUC values in the dataset. Whether this phenomenon is related to structural variations among athletes and HCM patients beyond differences in cardiac volumes should be further investigated. 

### 4.4. Limitations

This was a comparative experiment performed on two separate groups of individuals in whom different myocardial composition is certain. The results are not directly applicable to other populations. All comparative analysis should be interpreted in the context of the limited sample size. Despite the constraint of a limited sample size, the appearance of significant differences between the compared groups suggests the presence of important underlying diagnostic signals. These comparisons are merely hypothesis-generating for future studies. Finally, the maximal LV wall thickness is a common way of assessing hypertrophy, but the minor differences in wall thickness in our study were due to the study design and inclusion criteria.

## 5. Conclusions

Our study reveals that LV contraction heterogeneity in mild HCM is characterized by more pronounced mechanical dispersion as evidenced by strain echocardiography. Additionally, T1 mapping techniques by means of CMR demonstrate greater disarray and fibrosis in mild HCM patients than in athletes. These findings underscore the potential of speckle tracking echocardiography and CMR as sensitive tools for detecting myocardial alterations that can effectively differentiate between athletic cardiac remodeling and mild HCM. Our findings offer valuable insights into the mechanisms of cardiac remodeling, disclosing differences in physiological and pathological hypertrophy. 

Future research should evaluate the utility of speckle tracking echocardiography and CMR in broader populations, particularly in athletes presenting with suspected HCM. The identification of these differences could possibly also be applied in novel diagnostic pathways not only for HCM, but also for other cardiomyopathies. 

## Figures and Tables

**Figure 1 biomedicines-12-00420-f001:**
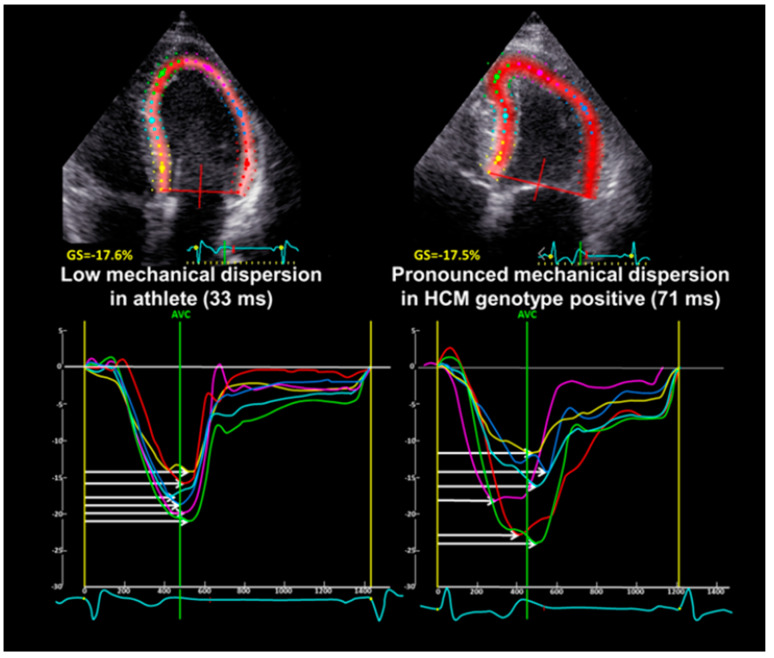
Echocardiographic strain analysis of an athlete (**left**) and mutation-positive HCM with a mild phenotype (**right**) obtained from the apical 4-chamber view. Mechanical dispersion was more pronounced in the mutation-positive HCM with a mild phenotype. White horizontal arrows represent segmental time from the onset of QRS to peak negative strain. AVC = aortic valve closure. HCM = hypertrophic cardiomyopathy.

**Figure 2 biomedicines-12-00420-f002:**
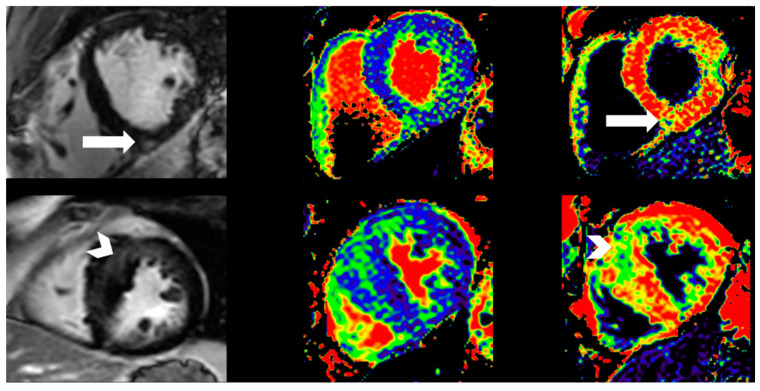
Contrast-enhanced cardiac magnetic resonance (**left**), native T1 map (**middle**) and post contrast T1 map (**right**) in an athlete (**top row**) and HCM (**lower row**) shown in short-axis views. Overall, HCM patients had more LGE and higher native T1 time than athletes. The athletic individual had one LGE focus at the lower RV insertion point into the septum (white arrow) with a local ECV of 36%, whereas the ECV in the normal septum was 21%. The patient with HCM had a broader, less sharply demarcated septal zone of LGE (white arrowhead) with an ECV of 31%, while the ECV in the remote, lateral wall was 21%. HCM = hypertrophic cardiomyopathy, ECV = extracellular volume fraction. LGE = late gadolinium enhancement. RV = right ventricular.

**Figure 3 biomedicines-12-00420-f003:**
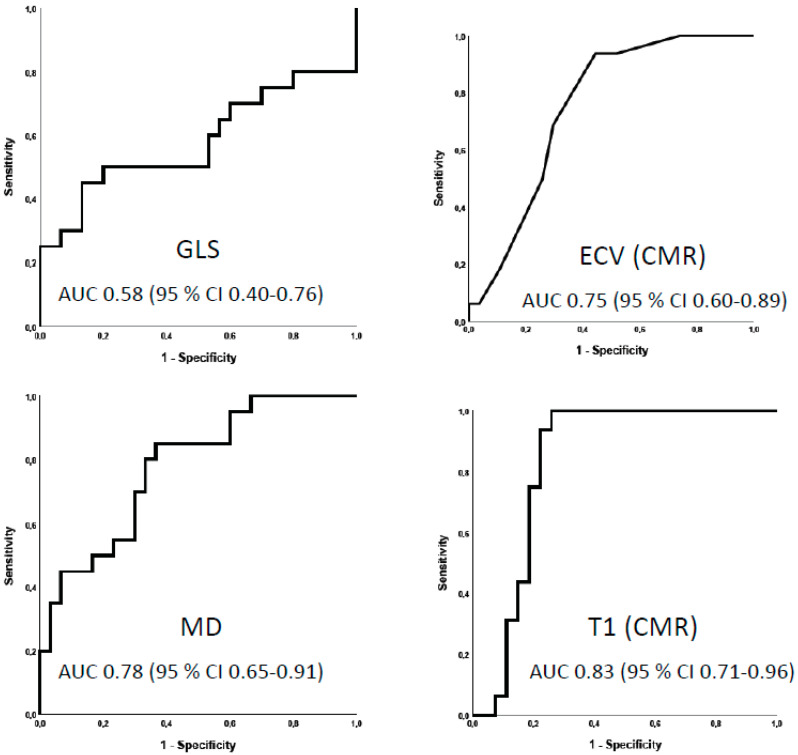
ROC curves of indices from strain echocardiography (**left** column) and T1 mapping techniques (**right** column). GLS > −17.7%, MD > 44 ms, T1 > 1230 ms and ECV > 22.5% were the optimal cut off values in the current dataset. CI = confidence interval. CMR = cardiac magnetic resonance imaging. ECV = extracellular volume. GLS = global longitudinal strain. MD = mechanical dispersion. T1 = native T1 time.

**Table 1 biomedicines-12-00420-t001:** Clinical characteristics and cardiac imaging.

	Healthy Athletes (n = 30)	Genopositive, Mild Phenotypic HCM (n = 20)	*p*
**Clinical**			
Age, years	41 ± 9	51 ± 12	0.002
BP, systolic, mmHg	122 ± 12	124 ± 17	0.651
BP, diastolic, mmHg	70 ± 11	74 ± 15	0.402
BSA, m^2^	2.0 ± 0.1	2.1 ± 0.2	0.04
Men/women, n	28/2	17/3	0.38
**Echocardiography**			
E/A ratio	1.6 ± 0.5	1.2 ± 0.4	0.003
E/e’ ratio	5.5 ± 1.2	9.8 ± 4.6	<0.001
e’, cm/s	11.4 ± 2.2	7.1 ± 2.0	<0.001
LA volume, mL/m^2^	44 ± 11	45 ± 16	0.77
LV 3D ejection fraction, %	55 ± 5	57 ± 6	0.16
LV 3D end-diastolic volume, mL/m^2^	90 ± 16	56 ± 9	<0.001
LV 3D mass, g/m^2^	66 ± 7	62 ± 9	0.13
LV global longitudinal strain, %	−18.9 ± 1.8	−18.1 ± 3.7	0.28
LV mechanical dispersion, ms	40 ± 11	54 ± 16	0.001
Maximal wall thickness, mm	10 ± 2	13 ± 1	<0.001
RV Fractional area change, %	39 ± 9	47 ± 5	0.001
**CMR**			
Extracellular volume, %	22.7 ± 3.2	25.6 ± 4.1	0.013
Late gadolinium enhancement, n (%)	1 (3)	8 (40)	0.001
LV ejection fraction, %	57 ± 6	62 ± 6	0.01
LV end-diastolic volume, mL/m^2^	114 ± 16	71 ± 12	<0.001
Native T1 time, ms	1204 (1191, 1234)	1265 (1255, 1312)	<0.001
RV ejection fraction, %	52 ± 6	61 ± 7	<0.001
RV end-diastolic volume, mL/m^2^	120 ± 18	72 ± 13	<0.001

Values are mean ± SD, median with the IQR, or n (%), and were compared using Student’s *t*-test, the Mann–Whitney U-test, χ^2^ test or Fischer’s exact test as appropriate. 3D = three-dimensional. A = late diastolic mitral flow velocity. BP = blood pressure. BSA = body surface area. CMR = cardiac magnetic resonance imaging. E = early diastolic mitral inflow velocity. e’ = early diastolic mitral annular tissue velocity. LA = left atrial. LV = left ventricular. RV = right ventricular.

**Table 2 biomedicines-12-00420-t002:** Summary of results.

	Athletes	HCM	*p*	Optimal Cut Off	AUC	95% CI
**Echocardiography**						
Mechanical dispersion, ms	40 ± 11	54 ± 16	0.001	>44	0.78	0.65–0.91
Global longitudinal strain, %	−18.9 ± 1.8	−18.1 ± 3.7	0.28	>−17.7	0.58	0.40–0.76
**CMR**						
Extracellular volume, %	22.7 ± 3.2	25.6 ± 4.1	0.013	>22.5	0.75	0.60–0.89
Native T1 time, ms	1204 (1191, 1234)	1265 (1255, 1312)	<0.001	>1230	0.83	0.71–0.96

Values are mean ± SD or median with the IQR and were compared using Student’s *t*-test or the Mann–Whitney U-test as appropriate. Indices from strain echocardiography and CMR and their ability to identify HCM mutation carriers from athletes in the study population were evaluated by means of ROC analysis. AUC = area under the curve. CI = confidence interval. CMR = cardiac magnetic resonance imaging. HCM = hypertrophic cardiomyopathy.

## Data Availability

Data are contained within the article and Appendix A.

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
