# Peer review of "Differentiation of Myocardial Properties in Physiological Athletic Cardiac Remodeling and Mild Hypertrophic Cardiomyopathy"

_biomedicines, 2024, doi:10.3390/biomedicines12020420_

Round 1

Reviewer 1 Report

Comments and Suggestions for Authors

The authors compared results from echocardiography and cardiac magnetic resonance imaging in patients with HCM compared to a healthy athletic population. They identified parameters in both echocardiography and cardiac magnetic resonance imaging that could distinguish between both entities: native T1-time (from MRI) and mechanical dispersion (from echo).

In my opinion, this is a very important paper that gives hints in the evaluation in patients with HCM that may also be active. It is well written with adequate statistics and references.

A table summarizing the main differences and thresholds to distinguish between HCM and athletic patients “in a nutshell” may be favourable. I have no other comments.

Author Response

The authors compared results from echocardiography and cardiac magnetic resonance imaging in patients with HCM compared to a healthy athletic population. They identified parameters in both echocardiography and cardiac magnetic resonance imaging that could distinguish between both entities: native T1-time (from MRI) and mechanical dispersion (from echo).

In my opinion, this is a very important paper that gives hints in the evaluation in patients with HCM that may also be active. It is well written with adequate statistics and references.

A table summarizing the main differences and thresholds to distinguish between HCM and athletic patients “in a nutshell” may be favorable. I have no other comments.

We thank Reviewer #1 for constructive and important comments. We have included a new table summarizing the parameters of interest from strain echocardiography and CMR with results from ROC analysis in line 201-206 of the revised manuscript.

Table 2 Summary of results

Athletes

HCM

P

Optimal cut off

AUC

95 % CI

Echocardiography

Mechanical dispersion, ms

40 ± 11

54 ± 16

0.001

> 44

0.78

0.65-0.91

Global longitudinal strain, %

-18.9 ± 1.8

-18.1 ± 3.7

0.28

> -17.7

0.58

0.40-0.76

CMR

Extracellular volume, %

22.7 ± 3.2

25.6 ± 4.1

0.013

> 22.5

0.75

0.60-0.89

Native T1 time, ms

1204 (1191, 1234)

1265 (1255, 1312)

<0.001

> 1230

0.83

0.71-0.96

Values are mean ± SD or median with IQR and compared by Students t-test or Mann Whitney U-test as appropriate. Indices from strain echocardiography and CMR and their ability to identify HCM mutation carriers from athletes in the study population evaluated by ROC analysis. AUC = Area under the curve. CI = Confidence interval. CMR = cardiac magnetic resonance imaging. HCM = Hypertrophic cardiomyopathy.

Reviewer 2 Report

Comments and Suggestions for Authors

The study entitled "Differentiating Myocardial Properties in Physiological Athletic Cardiac Remodeling and Mild Hypertrophic Cardiomyopathy" aims to clarify the differences between mutation-positive mild phenotypic hypertrophic cardiomyopathy (HCM) and the athletic heart. To this end, strain echocardiography and cardiac magnetic resonance (CMR) indices will be used to evaluate their effectiveness in differentiating between these two conditions.

The research results show that the heterogeneity of left ventricular contraction in mild HCM can be more clearly identified by improved mechanical dispersion observed in strain echocardiography. In addition, T1 mapping techniques in CMR show a higher degree of disarray and fibrosis in mild HCM compared to athletic hearts. These results provide valuable insights into the mechanisms of cardiac remodeling and may pave the way for new diagnostic approaches to distinguish between physiological and pathological hypertrophy.

Author Response

The study entitled “Differentiating Myocardial Properties in Physiological Athletic Cardiac Remodeling and Mild Hypertrophic Cardiomyopathy” aims to clarify the differences between mutation-positive mild phenotypic hypertrophic cardiomyopathy (HCM) and the athletic heart. To this end, strain echocardiography and cardiac magnetic resonance (CMR) indices will be used to evaluate their effectiveness in differentiating between these two conditions.

The research results show that the heterogeneity of left ventricular contraction in mild HCM can be more clearly identified by improved mechanical dispersion observed in strain echocardiography. In addition, T1 mapping techniques in CMR show a higher degree of disarray and fibrosis in mild HCM compared to athletic hearts. These results provide valuable insights into the mechanisms of cardiac remodeling and may pave the way for new diagnostic approaches to distinguish between physiological and pathological hypertrophy.

We would like to thank the reviewer for these good comments. Two new references and a table summarizing the main results are now included in the revised manuscript.

Reviewer 3 Report

Comments and Suggestions for Authors

The article submitted for review is of interest to me in routine clinical practice as the differentiation of exercise-induced myocardial hypertrophy from hypertrophic cardiomyopathy is sometimes difficult.

The number of patients is small and an adequate sample size calculation is not available. I think the authors should be more precise in this respect.

Author Response

The article submitted for review is of interest to me in routine clinical practice as the differentiation of exercise-induced myocardial hypertrophy from hypertrophic cardiomyopathy is sometimes difficult.

The number of patients is small and an adequate sample size calculation is not available. I think the authors should be more precise in this respect.

We appreciate your insightful comments regarding the small sample size limitation in our study. We recognize that a small sample size may compromise the statistical power to discern differences between groups. However, it is important to note that our study involved a highly selected cohort of participants recruited under stringent criteria to mitigate confounding factors associated with cardiac remodeling. While the sample size indeed was limited, the statistical differences observed in our study, indicate that it was adequate to detect the diagnostic signal of interest. To further elucidate this point, we have added the following clarification in the limitation section (lines 301-302).

Despite the constraint of a limited sample size, the appearance of significant differences between the compared groups suggests the presence of important underlying diagnostic signals.

Reviewer 4 Report

Comments and Suggestions for Authors

I would congratulate with authors for this brilliant paper reporting that LV contraction heterogeneity in mild HCM can be identified by more pronounced mechanical dispersion using strain echocardiography, and T1 mapping techniques indicates more disarray and fibrosis in mild HCM than in athletes. Paper is extremely good, I have only one minor comment in order to improve the manuscript. In discussion author’s should cite the impact of sport both in athlete’s heart as well as mild HCM, specially in terms of progression (DOI: 10.1016/j.ijcard.2021.10.013  ;  DOI: 10.1093/eurjpc/zwad011). Please expand the concepts and cite two suggested references

Author Response

I would congratulate with authors for this brilliant paper reporting that LV contraction heterogeneity in mild HCM can be identified by more pronounced mechanical dispersion using strain echocardiography, and T1 mapping techniques indicates more disarray and fibrosis in mild HCM than in athletes. Paper is extremely good, I have only one minor comment in order to improve the manuscript.

In discussion author’s should cite the impact of sport both in athlete’s heart as well as mild HCM, specially in terms of progression (DOI: 10.1016/j.ijcard.2021.10.013  ;  DOI: 10.1093/eurjpc/zwad011). Please expand the concepts and cite two suggested references

We sincerely thank the Reviewer 4 for generous and important comments. We agree that these papers should be cited. Accordingly, we added the following changes to the discussion in line 227-229 and cited the above-mentioned papers in the revised manuscript.

4.1. Physiological and pathological myocardial remodeling

Undiagnosed underlying heart disease is the leading cause of sudden cardiac death in athletes. In younger athletes, inherited cardiomyopathies such as HCM are the main causes of these tragic events [20]. Traditionally, people with HCM have been advised not to participate in sports. This view has become more nuanced in recent years as data regarding athletic cardiac remodeling and impact of physical activity in HCM is accumulating [21, 22].

Round 2

Reviewer 3 Report

Comments and Suggestions for Authors

The article submitted for review deals with the interesting topic of finding new contributions in diagnostic techniques for the study of ventricular remodeling. 

The design is adequate, but in my opinion the conclusions section should be expanded as there are many conclusions and future lines of study that this article can provide. With these modifications, I would value the article again for publication in the journal. 

Author Response

Reviewer 3

The article submitted for review deals with the interesting topic of finding new contributions in diagnostic techniques for the study of ventricular remodeling. 

The design is adequate, but in my opinion the conclusions section should be expanded as there are many conclusions and future lines of study that this article can provide. With these modifications, I would value the article again for publication in the journal. 

We appreciate your remarks regarding the conclusion. In response to your suggestion, we have expanded the conclusion by incorporating methodological considerations and outlining potential future lines of study. This addition aims to provide a more comprehensive overview of the implications of our findings. The following text was added to the conclusion in the revised manuscript:

  1. Conclusions

Our study reveals that LV contraction heterogeneity in mild HCM can be identified is characterized by more pronounced mechanical dispersion as evidenced using by strain echocardiography. and Additionally, T1 mapping techniques by CMR demonstrates indicates more greater disarray and fibrosis in mild HCM than in athletes. These findings underscore the potential of speckle tracking echocardiography and CMR as sensitive tools for detecting myocardial alterations that can effectively differentiate between athletic cardiac remodeling and mild HCM. Our findings give offer valuable insight into mechanisms of cardiac remodeling, and might disclosing differences in new diagnostic pathways to differentiate between physiological and pathological hypertrophy.

Future research should evaluate the utility of speckle tracking echocardiography and CMR in broader populations, particularly in athletes presenting with suspected HCM. The identification of these differences could possibly also be applied in novel diagnostic pathways not only for HCM, but also for other cardiomyopathies. The value of multimodality imaging in this clinical scenario should be explored in future studies.
